# CircRNAome of Childhood Acute Lymphoblastic Leukemia: Deciphering Subtype-Specific Expression Profiles and Involvement in *TCF3::PBX1* ALL

**DOI:** 10.3390/ijms25031477

**Published:** 2024-01-25

**Authors:** Angela Gutierrez-Camino, Maxime Caron, Chantal Richer, Claire Fuchs, Unai Illarregi, Lucas Poncelet, Pascal St-Onge, Alain R. Bataille, Pascal Tremblay-Dauphinais, Elixabet Lopez-Lopez, Mireia Camos, Manuel Ramirez-Orellana, Itziar Astigarraga, Éric Lécuyer, Guillaume Bourque, Idoia Martin-Guerrero, Daniel Sinnett

**Affiliations:** 1Division of Hematology-Oncology, CHU Sainte-Justine Research Center, Montreal, QC H3T 1C5, Canada; angela.gutierrez@ehu.eus (A.G.-C.); max.caron@mail.mcgill.ca (M.C.); chantal.richer.hsj@ssss.gouv.qc.ca (C.R.); claire.fuchs.hsj@ssss.gouv.qc.ca (C.F.); lucas.poncelet@cnrc-nrc.gc.ca (L.P.); pascal.st-onge@umontreal.ca (P.S.-O.); alain.bataille.hsj@ssss.gouv.qc.ca (A.R.B.); pascal.tremblay-dauphinais.hsj@ssss.gouv.qc.ca (P.T.-D.); 2Department of Human Genetics, McGill University, Montreal, QC H3A 0G4, Canada; guil.bourque@mcgill.ca; 3Department of Genetics, Physical Anthropology and Animal Physiology, Faculty of Science and Technology, University of the Basque Country (UPV/EHU), 48940 Leioa, Spain; unai.illarregi@ehu.eus (U.I.); idoia.marting@ehu.eus (I.M.-G.); 4Department of Biochemistry and Molecular Biology, Faculty of Science and Technology, University of the Basque Country (UPV/EHU), 48940 Leioa, Spain; elixabet.lopez@ehu.eus; 5Pediatric Oncology Group, Biobizkaia Health Research Institute, 48903 Barakaldo, Spain; mariaiciar.astigarragaaguirre@osakidetza.eus; 6Hematology Laboratory, Sant Joan de Déu Research Institute, Esplugues de Llobregat, 08950 Barcelona, Spain; mireia.camos@sjd.es; 7Department of Pediatric Hematology and Oncology, Niño Jesús University Hospital, 28009 Madrid, Spain; manuel.ramirez@salud.madrid.org; 8Department of Pediatrics, Cruces University Hospital, 48903 Barakaldo, Spain; 9Institut de Recherches Cliniques de Montréal (IRCM), Montreal, QC H2W 1R7, Canada; eric.lecuyer@ircm.qc.ca; 10Département de Biochimie et Médecine Moléculaire, Université de Montréal, Montreal, QC H3C 3J7, Canada; 11Division of Experimental Medicine, McGill University, Montreal, QC H3A 0G4, Canada; 12Department of Pediatrics, Faculty of Medicine, University of Montreal, Montreal, QC H3C 3J7, Canada

**Keywords:** acute lymphoblastic leukemia, childhood B-ALL, circular RNA, *TCF3::PBX1*, RNA-binding protein, *NUDT21*

## Abstract

Childhood B-cell acute lymphoblastic leukemia (B-ALL) is a heterogeneous disease comprising multiple molecular subgroups with subtype-specific expression profiles. Recently, a new type of ncRNA, termed circular RNA (circRNA), has emerged as a promising biomarker in cancer, but little is known about their role in childhood B-ALL. Here, through RNA-seq analysis in 105 childhood B-ALL patients comprising six genetic subtypes and seven B-cell controls from two independent cohorts we demonstrated that circRNAs properly stratified B-ALL subtypes. By differential expression analysis of each subtype vs. controls, 156 overexpressed and 134 underexpressed circRNAs were identified consistently in at least one subtype, most of them with subtype-specific expression. *TCF3::PBX1* subtype was the one with the highest number of unique and overexpressed circRNAs, and the circRNA signature could effectively discriminate new patients with *TCF3::PBX1* subtype from others. Our results indicated that *NUDT21*, an RNA-binding protein (RBP) involved in circRNA biogenesis, may contribute to this circRNA enrichment in *TCF3::PBX1* ALL. Further functional characterization using the CRISPR-Cas13d system demonstrated that *circBARD1*, overexpressed in *TCF3::PBX1* patients and regulated by *NUDT21*, might be involved in leukemogenesis through the activation of p38 via *hsa-miR-153-5p*. Our results suggest that circRNAs could play a role in the pathogenesis of childhood B-ALL.

## 1. Introduction

Childhood acute lymphoblastic leukemia (ALL), the most frequent pediatric cancer [1], is a heterogeneous disease comprising multiple molecular subtypes with distinct somatic genetic alterations, such as aneuploidy or chromosomal rearrangements [1,2], associated with various clinical outcomes. Hyperdiploid cases and patients harboring the *ETV6::RUNX1* rearrangement together represent about half of B-ALL cases and are associated with good prognosis [1,3]. By contrast, less frequent (<10%) subtypes, such as *MLL*-rearranged or *BCR::ABL1*, are associated with less favorable outcomes [1] and a higher risk of relapse. These genetic lesions determine subtype-specific expression profiles of coding genes, as well as non-coding transcripts such as miRNAs and lncRNAs [4,5,6]. Recently, a new type of ncRNA, termed circular RNA (circRNA), has emerged as a promising biomarker in cancer.

CircRNAs are closed RNA molecules mainly expressed from protein coding genes (referred as parental genes) by a process called backsplicing [7,8]. Backsplicing of circRNAs can be regulated by the direct base pairing between inverted repeat elements in the flanking intron sequences (such as *Alu* elements) and the binding of RNA-binding proteins (RBPs). For instance, RBPs such as *QKI* and *FUS*, drive exon circularization through their binding of specific motifs in the flanking introns [9,10,11], while *ADAR1* and *DHX9* suppress circRNAs biogenesis [10,12]. The most described roles of circRNAs include the regulation of gene expression through the sequestration of miRNAs by acting as a “sponge” protecting target mRNAs from miRNA-dependent degradation [8]. circRNAs containing RBP binding motifs may function as sponges or decoys and indirectly regulate their functions [11,13]. Moreover, circRNAs may function as protein scaffolds and recruiters or by being translated themselves into small peptides [8]. RNA-seq studies showed that circRNAs are highly expressed in a tissue-specific and cell-type-specific manner [14]. This specificity has been demonstrated in hematopoietic cells, with dynamic regulation through maturation and differentiation processes [7,15]. CircRNA expression levels often correlate with clinical and pathological characteristics, particularly in certain types of leukemias [16,17,18]. Recent studies identified differentially expressed circRNAs with specific roles in cancer [19,20]. Regarding childhood ALL, Buratin et al. [21] observed a deregulation of the circRNAome in T-ALL patients as well as subtype-specific expression. Moreover, they identified an oncogenic role for *circZNF609*, which showed a decrease in cell viability upon silencing. Gaffo et al. [22] demonstrated the deregulation of up to four circRNAs in different B-ALL subtypes. Although those results point to the functional involvement of circRNAs in childhood ALL, little is known about the specific profiles and roles of circRNAs in the development and treatment of childhood B-ALL. The aim of this study was to characterize circRNA expression profiles using an RNA-seq approach and to investigate their role in childhood B-ALL pathogenesis.

## 2. Results

### 2.1. Characterization of the circRNAome in Childhood B-ALL

To investigate the full spectrum of circRNAs expressed during malignant B-cell development, we examined high-depth ribo-depleted RNA-seq data of 105 B-ALL patients representing six genetic B-ALL subtypes: *ETV6::RUNX1*/like, Hyperdiploid, *DUX4*-rearranged, *MLL*-rearranged, Ph+/like, and *TCF3::PBX1* (Table 1). Although the discovery and replication cohorts contained different numbers of patients, all subtypes were well represented, except for *MLL*-R in the replication cohort. The age and sex distributions of the patients were similar in both cohorts. As controls, CD10+/CD19+ B cells purified from human cord blood in the discovery cohort and CD19+ B cells purified from the peripheral blood of healthy children in the validation cohort were included. Peripheral blood represents mature B cells. Because circRNAs are cell-type specific and these differences could bias our results, we determined the predominant hematopoietic cell signal within each bulk transcriptome by deconvolution. We observed that both groups predominantly classified as immature cells (CD34−/CD19+/IgM+) (Appendix A), indicating a similar transcriptional maturation profile. This result prompted us to study the maturation states of leukemia transcriptomes, identifying mainly pro-B profiles (CD34−/CD19+/IgM−) (Appendix A). Recent data suggest that the developmental stage of B cells could significantly affect the transcriptional landscape of cells [23]; therefore, some circRNAs identified in our analysis might reflect differences between pro-B and immature cells. To mitigate potential bias in circRNA detection, both cohorts were treated separately and only results observed in both populations were included in our circRNA profiles, increasing the reproducibility of our results.

Quantification and annotation of circRNAs in B-ALL patients and controls resulted in the identification of 245,887 circRNAs in total, including 83,944 minimally expressed (≥1 backsplice junction reads in ≥3 samples). To generate a highly reliable list of circRNA, datasets enriched for circRNA were generated from five leukemia and one normal B-cell line by degrading linear RNA and depleting poly(A)-containing RNAs. The resulting profiles were then compared with those from the patients. Only circRNAs that were detected in both experiments were considered highly confident circRNAs and were retained for further analysis, resulting in a final count of 40,771 circRNAs (Figure 1A), including 13,332 new candidates that were not annotated in circBase. The circRNAs identified in both cohorts were similar in size (~600 bp), number of exons per circRNA (~4), and number of circRNA isoforms per gene (~5) (Figure 1B). Moreover, CIRCscore, the ratio between circular and linear RNA expression, indicated that around 2% of the circRNAs in both cohorts were more expressed than their parental gene (CIRCscore > 1) (Appendix A). The number of circRNAs identified was comparable across subtypes of patients in both cohorts, except for the *ETV6::RUNX1* patients and B controls (Figure 1C). The increased number of circRNAs in the discovery cohort for *ETV::RUNX1* ALL patients may be explained by the larger number of patients (31 vs. 7). In the B controls, which had a similar number of samples, the lower number of circRNAs in the discovery cohort may be attributed to the difference in coverage, with 108× and 178× in the discovery and replication cohorts, respectively.

### 2.2. Subtype-Specific Profile of circRNAs in Childhood B-ALL

Unsupervised clustering of both cohorts based on the most variable genes or circRNAs revealed subtype specific clusters (Figure 2A), indicating that circRNA expression effectively stratifies childhood B-ALL subtypes. Subtype-specific circRNAs were identified using differential expression analyses comparing patients to controls (Figure 2B). In the discovery cohort, 553 differentially expressed (DE) circRNAs were found (*p*-adj < 0.05), of which 325 were overexpressed (58.8%) and 228 were downregulated (41.2%). In the replication cohort, we found 836 overexpressed (36%) and 1506 downregulated circRNAs (64%) (Appendix A). Integrated analysis showed a total of 156 overexpressed and 134 underexpressed unique circRNAs (*p*-adj < 0.05) (Appendix A). Notably, *TCF3::PBX1* was the only subtype with a clear upregulation of circRNA expression, displaying the highest number of overexpressed circRNAs (n = 82). The *TCF3::PBX1* subtype also had the highest number of unique overexpressed circRNAs (n = 61) (Figure 2C). Moreover, the circRNAs with the highest CIRCscore in patients and very low CIRCscore in controls were also found in patients carrying *TCF3::PBX1* translocation in both cohorts (Appendix A). Interestingly, Hyperdiploid and *MLL*-rearranged ALL subtypes shared up to five circRNAs, corresponding to different circular isoforms of FLT3 highly correlated with FLT3 linear expression and not annotated in circBase [24]. This result differs from the low correlation observed for different isoforms of circFLT3 in acute myeloid leukemia (AML) [17], which may be due to the different mechanisms of *FLT3* activation in both diseases (mutation vs. overexpression) [25]. Finally, no circRNA was found to be overexpressed in every B-ALL subtype, as opposed to the 10 circRNAs downregulated in the B-ALL patients (Figure 2D).

Functional predictions for the 156 upregulated circRNAs revealed that those found in the *TCF3::PBX1* subtype were enriched for miRNA binding sites and displayed higher coding potential (Appendix A). For example, *hsa_circ_0052318* (ZNF418) contains multiple binding sites for miR-199a and miR-199b, which are both underexpressed in childhood ALL [4]. By decreasing the availability of miRNAs miR-199a and miR-199b, the circRNA *hsa_circ_0052318* has the potential to increase the expression of important oncogenic targets in ALL, including STAT3, AKT, JUNB and KDM6A [26]. Another circRNA, *hsa_circ_0031584* (ARHGAP5), had some of the highest values for coding probability and m^6^A RNA modification scores, according to circBank. Moreover, it contained upstream internal ribosomal entry sites (IRES), an open reading frame (ORF), and protein features annotated in the circRNADB database.

Finally, the expression of circRNA was analyzed in association with patients’ characteristics and outcomes. Interestingly, several circRNAs were specifically overexpressed in boys (Appendix A) while no other associations were found with age, relapse, or survival.

### 2.3. CircRNAome in TCF3::PBX1 B-ALL

To further investigate the role of circRNAs in the *TCF3::PBX1* subtype, six new patients carrying this translocation were sequenced and included in the study as part of the discovery cohort. Hierarchical clustering using the 156 overexpressed and 134 underexpressed circRNAs previously identified revealed that our circRNA signature could effectively discriminate new patients with the *TCF3::PBX1* subtype from others (Figure 3A). Moreover, differential expression analysis between *TCF3::PBX1* and other B-ALL subtypes demonstrated that even more circRNAs were deregulated in *TCF3::PBX1* patients (Appendix A).

This specific involvement of circRNAs in *TCF3::PBX1* pathogenesis might be due to alterations in circRNA biogenesis. Recent studies suggested that a broad range of RBPs are involved in circRNA formation [9,10,11,12,27] and alterations in these genes can lead to circRNA dysregulation. To explore whether the deregulation of RBPs affects circRNA profiles, the expression levels of 10 genes involved in circRNA biogenesis were analyzed using our RNA-seq data. Interestingly, the *TCF3::PBX1* subgroup was the only one showing an overexpression of *NUDT21* in comparison to controls in both cohorts (Figure 3B and Appendix A). Since *NUDT21* promotes the formation of circRNAs [27], we hypothesized that this RBP could be involved in the specific overexpression of circRNAs in *TCF3::PBX1* ALL. Indeed, our analysis showed a correlation between *NUDT21* expression levels and the *TCF3::PBX1* subtype-specific circRNAs (Pearson’s correlation = 0.55, *p*-value = 0.04) (Appendix A). To further explore the role of NUDT21 in circRNA biogenesis, we used the CRISPR-Cas13d system to silence *NUDT21* in 697 cells and analyzed the expression levels of 10 circRNA candidates and their corresponding parental genes by RT-qPCR. The candidates were selected based on the overlap between circRNAs upregulated in *TCF3::PBX1* patients and those expressed in the 697 cell line and ranked according to their expression in the cell line. Our results showed that for 3 out of the 10 candidates (*circANKS1B*, *circBARD1*, and *circMAN1A2*), silencing *NUDT21* barely affected the expression of the linear forms while it decreased circRNA expression by 40% (Figure 3C). These results point to a role of *NUDT21* in the overexpression of circRNAs observed in *TCF3::PBX1* ALL.

### 2.4. Functional Validation of circBARD1

*CircBARD1* (*hsa_circ_0001098*), overexpressed in *TCF3::PBX1* patients and downregulated under *NUDT21* knockdown conditions, was selected for functional validation due to its high expression in patients, 697 cell line, and functional predictions (Appendix A). By using the CRISPR-Cas13d system for loss-of-function studies, we reached a silencing efficiency of 83% without affecting the linear mRNA of the parental gene (Figure 4A). We then assessed the impact of *circBARD1* knockdown (Cas13d-gcircBARD1) on cancer phenotypes including cell proliferation, migration, and oxidative stress response compared to control cells (Cas13d-NT) stably expressing Cas13d and a non-targeting gRNA. Cells expressing *circBARD1* presented a higher proliferation rate and migration capabilities than the silenced cells (*p* < 0.05) (Figure 4B,C). Interestingly, *circBARD1* expression resulted in higher sensitivity to increasing concentrations of H_2_O_2_ levels (Figure 4D), a reactive oxygen molecule (ROS). Accumulating evidence suggests that cancer cells tend to show higher amounts of ROS to maintain tumorigenic conditions [28], but also that ROS levels are a “double-edged sword” making cancer cells more susceptible to H_2_O_2_-induced cell death [29], due to their decreased ability to metabolize H_2_O_2_ [29,30].

We next investigated the mechanism of action of *circBARD1* focusing on miRNA binding capacity. Using cell fractionation and RT-qPCR, we determined that *circBARD1* was mainly enriched in the cytoplasm (Figure 4E), supporting its role as miRNA sponge. Functional predictions from circBank and TargetScan pointed to *hsa-miR-153-5p* as the primary candidate for *circBARD1* binding. We retrieved a total of 211 experimentally validated miRNA-target interactions from miRTarbase for *hsa-mir-153-5p* and performed pathway enrichment analysis, which showed p38 MAPK signaling over-represented among these targets, with four candidate genes: *MAKP14* (p38), *TAOK3*, *MAP3K1*, and *MAP3K7*. We quantified the expression levels of these genes in *TCF3::PBX1* patients and controls and found that *MAPK14* (p38) was overexpressed in *TCF3::PBX1* ALL in both cohorts (Appendix A). Furthermore, the interaction of *circBARD1* and p38 was also supported by protein phosphorylation experiments that showed a decreased p38 phosphorylation in *circBARD1* knockdown conditions compared to controls in 697 cells (Figure 4F). We also observed a lower phosphorylation of STAT3 protein (Figure 4F). Collectively, our data suggest that *circBARD1* plays a role in the pathogenesis of *TCF3::PBX1* ALL.

## 3. Discussion

This study aimed to characterize the circRNAomes of 105 B-ALL children including six molecular subtypes and seven B-cell controls from two independent cohorts. To enhance the reproducibility of our findings, both cohorts were treated separately to reduce potential bias in circRNA detection due to population characteristics, library preparation, and sequencing coverage. Moreover, circRNA-enriched datasets were generated from six leukemia and normal B-cell lines to reduce the false positive detection rate, identifying highly reliable circRNAs. A total of 40,771 validated circRNAs were identified in childhood B-ALL, including 13,332 novel circRNAs. Unsupervised clustering of the most variable circRNAs revealed subtype-specific clusters, demonstrating that circRNAs can accurately stratify B-ALL subtypes, extending the list of non-coding RNAs with subtype-specific expression in childhood ALL. We showed that 290 unique circRNA are consistently differentially expressed for at least one subtype in both cohorts, with 156 being overexpressed and 134 being underexpressed.

Focusing on overexpressed circRNAs, we found that most of them have subtype-specific expression, and half of these circRNAs were found in the *TCF3::PBX1* subtype, with 61 unique circRNAs. This subtype also had the highest number of overexpressed circRNAs and the highest CIRCscores despite having a low number of patients and similar sequencing characteristics to other subtypes. Importantly, the identified signature was able to differentiate *TCF3::PBX1* ALL from the rest of the subtypes, suggesting a specific involvement of circRNAs in *TCF3::PBX1* pathogenesis. To gain further insight into the biology of circRNAs in childhood B-ALL, we analyzed the expression levels of 10 RBPs involved in the formation of circRNAs [9,10,11,12,27]. Interestingly, *TCF3::PBX1* was the only subtype with a clear upregulation of one of the RBPs analyzed, *NUDT21*, a conserved splicing factor active in pre-mRNA assembly [31] as well as the formation of circRNAs [27]. Silencing of *NUDT21* reduced the expression levels of three out of ten circRNAs overexpressed in *TCF3::PBX1* patients without affecting their linear forms. Considering the specific overexpression profile of circRNAs observed in *TCF3::PBX1* patients, *NUDT21* may contribute to circRNA dysregulation in this subtype of B-ALL. Moreover, Witkowski et al. [32] recently demonstrated that loss of *NUDT21* has an important role in 697 cells, causing changes in gene expression and affecting cell viability. Indeed, enhanced crosslinking precipitation (eCLIP) of NUDT21-bound mRNA in 697 cells showed NUDT21-binding peaks across the linear forms of three candidates identified in the current study [32]. This binding enrichment could be due to its involvement in circRNA formation. Interestingly, *circBARD1* and *circMAN1A2*, which showed the highest downregulation under *NUDT21*-knockdown conditions, were also regulated by *NUDT21* in hepatocellular carcinoma cells [27].

To confirm the role of circRNAs in *TCF3::PBX1* ALL, *circBARD1* was selected for loss-of-function studies using the CRISPR/Cas13d technique, which demonstrated higher efficiency and lower off-target rates on linear mRNAs compared to shRNA-mediated circRNA silencing [33]. We found that *circBARD1* is mainly expressed in the cytoplasm and is involved in the proliferation, migration, and oxidative stress response. Moreover, cells expressing *circBARD1* showed a higher phosphorylation of p38 and STAT3 proteins, suggesting a link between the phenotypes observed upon silencing and key signaling pathways. We hypothesized that *circBARD1* activates the p38 MAPK signaling pathway via *hsa-miR-153-5p*. Overexpression of *circBARD1* in *TCF3::PBX1* patients could sequester *hsa-miR-153-5p*, consequently contributing to the activation of p38. Indeed, p38 expression levels are upregulated in patients with *TCF3::PBX1* ALL. While p38 activation is typically associated with anti-proliferative functions, there are reports indicating that p38 can positively regulate proliferation, as well as migration, in hematopoietic cells and several cancer cell lines [34], both features observed in *circBARD1*-expressing cells. Regarding STAT3, activation of this protein markedly contributes to the generation of mature hematopoietic cells by affecting cellular growth, survival, and lineage specificity [35]. Indeed, Karvonen et al. [36] recently demonstrated that *TCF3::PBX1* cells sustain cell survival and proliferation via STAT3 signaling pathways. circBARD1 may contribute to this activation by other mechanisms, such as acting as an RBP sponge, which is another enriched functional prediction for this circRNA. Although further studies are needed to characterize the exact mechanism of action of *circBARD1* in 697 cells, as well as the involvement of this circRNA in other cell lines carrying *TCF3::PBX1* translocation, collectively, our data indicate that *circBARD1* is involved in *TCF3::PBX1* ALL.

The current study has some limitations that need to be addressed. For instance, some circRNAs identified might reflect differences between pro-B and immature B cells. Considering that circRNA expression in hematopoietic progenitors increases upon maturation [15], it is reasonable to expect that more mature cells will exhibit greater variability in circRNA species, and higher expression due to their enhanced stability. Since our controls are more mature B cells than leukemia samples, circRNAs downregulated in B-ALL might be indicative of the maturation stage. Moreover, considering that the six subtypes shared a similar expression profile with regard to maturation, the circRNAs identified in at least five subtypes are potential circRNA candidates associated with the lymphopoiesis process. More downregulated circRNAs were shared among subtypes, whereas most of the overexpressed circRNAs were subtype specific, indicating that upregulated circRNAs are likely associated with specific modification acquired by the genetic molecular background. Finally, only a few RNA-binding factors involved in circRNA biogenesis have been discovered [37]. Here, we focused on the involvement of RBPs, but recent findings suggest that m^6^A modifications [38] and epigenetic changes within histones and genes [8] may also have a direct impact on circRNA biogenesis. Further studies on the mechanisms of circRNA dysregulation could help elucidate the whole picture of aberrant circRNA expression observed in *TCF3::PBX1* patients.

In conclusion, we found 290 circRNAs deregulated in at least one B-ALL subtype, providing a list of high-confidence circRNAs with potential for functional characterization. We demonstrated that circRNAs exhibit subtype-specific expression in childhood ALL and that patients with *TCF3::PBX1* translocation are enriched in subtype-specific and overexpressed circRNAs. Overexpression of *NUDT21* in *TCF3::PBX1* patients may contribute to circRNA enrichment in this subtype of childhood B-ALL. Moreover, *circBARD1*, overexpressed in *TCF3::PBX1* patients and regulated by *NUDT21*, is involved in leukemogenesis through the activation of p38 via *hsa-miR-153-5p*. Our results suggest that circRNAs could play a role in the pathogenesis of this ALL subtype. The identification of new factors involved in childhood ALL etiology, progression, and treatment response will not only add a new layer of complexity in leukemogenesis but may also lead to the identification of new therapeutic targets.

## 4. Materials and Methods

### 4.1. Patient Samples

The discovery cohort consisted of 68 patients from the Quebec childhood ALL cohort (QcALL) diagnosed with childhood B-ALL in the Hematology–Oncology department at the Sainte-Justine Hospital (Montreal, QC, Canada) [39]. Three control samples were purified from human cord blood by sorting CD10+/CD19+ B cells [5]. The validation cohort consisted of 37 patients with childhood B-ALL diagnosed at the Pediatric Oncology Departments of Cruces University Hospital (Bilbao, Spain), Niño Jesús University Childhood Hospital (Madrid, Spain), and Sant Joan de Deu Hospital (Barcelona, Spain). Four control samples were purified from the peripheral blood of healthy children by sorting CD19+ B cells (Table 1). The use of different cohorts reduced the biological variability due to population characteristics and increased the reproducibility of results.

### 4.2. Library Preparation

RNA libraries (TruSeq Stranded Total RNA Library Prep Kit; Illumina, San Diego, CA, USA) were prepared using the Ribo-Zero Gold kit (Illumina) and according to the manufacturer’s protocol. The resulting libraries (stranded and ribosomal RNA depleted) were sequenced at approximately 150M reads per sample, paired end (2 × 75 bp; 2 × 100 bp) (Appendix A) on the HiSeq 4000 and NovaSeq 6000 System (Illumina). All sequencing was performed at the Integrated Center for Pediatric Clinical Genomics of the Sainte-Justine University Hospital Research Center.

### 4.3. Bioinformatic Analyses

To ensure that the subtype classification of the patients was homogeneous in both cohorts, we applied a leukemia subtype classification tool combining clinical data and coding gene expression, as previously described [40]. Briefly, we trained a neural network classifier on 195 published RNA-seq datasets [41] using the top 500 most statistically significant differentially expressed genes among 9 ALL subtypes. The trained model was applied to each patient’s sample in our study, and the subtype with the highest probability was reported as the predicted ALL subtype. To assess differences in transcriptome profiles arising from the isolation of B cells at different maturation states in the patient subtypes and controls, we determined the predominant hematopoietic cell signal within each bulk transcriptome by deconvolution using CIBERSORTx [42]. A public mRNA dataset (GSE115655) [43] containing profiles of early B (CD34+/CD19−/IgM−), pro-B (CD34−/CD19+/IgM−), preB (CD34+/CD19+/IgM−), and immature cells (CD34−/CD19+/IgM+) was used to build the reference signature matrix.

CircExplorer v3 [44] was used to quantify the expression of circRNAs and circBase [24] version 2017 was used for the annotation. CircRNAs with at least one backsplicing junction read in at least three samples were retained for differential expression analyses. Comparison of circular and linear RNA expression levels was carried out using the CIRCscore [44], a tool that evaluates the circular and linear RNA expression ratio individually using fragments mapped to circRNA-specific backsplice junction sites and the linear RNA-specific exon–exon splicing junction. A CIRCscore > 1 indicates circRNAs with a higher expression than their linear counterparts. Differential expression analyses between each subtype and control were performed using DESeq2 v1.16.1 [45]. CircRNAs with a p-adjusted value < 0.05 were selected for downstream analyses. Two computational tools, the oRNAment database [46] and circInteractome [47], were used to annotate RBP binding based on the circular sequence. MiRNA binding sites, m^6^A motifs, and coding potential were annotated using circBank database [48] and circRNADB [49]. For specific analysis on *circBARD1*, miRNA binding sites were predicted using TargetScan [50] (release 8.0, September 2021) and validated miRNA target genes were retrieved with miRTarbase [26] (release 9.0 beta: 15 September 2021). Pathway enrichment analyses were performed using ConsensusPathDB-human (release 35, June 2021). Fragments per kilobase per million mapped read (FPKM) values for genes of interest were obtained from the RNA-seq data using cufflinks v2.2.1 [51] on the Ensembl version 75 gene annotation and log transformed. Statistical analyses using FPKM were conducted using GraphPad Prism 5.0 (GraphPad Software, Boston, MA, USA). A *p*-value < 0.05 was considered statistically significant for all comparisons.

### 4.4. Quantification in Cell Lines

Total ribosomal-depleted RNA and circRNA-enriched transcriptomes from 5 leukemia cell lines (REH, RS4:11, SEM, NALM6, and 697) and the normal B-cell line GM12878 (American Type Culture Collection, Manassas, VA, USA; Coriell Institute for Medical Research, Camden, NJ, USA; DSMZ-German Collection of Microorganisms and Cell Cultures GmbH, Braunschweig, Germany) were sequenced on Hiseq 4000 System (Illumina). CircRNA enrichment was performed by RNase R treatment, polyadenylation, and poly(A)+ RNA depletion (RPAD) [52]. Overlapping circRNAs between the RPAD-treated cell lines and patient samples were defined as high-confident circRNAs and used for downstream analyses. Those overlapping circRNA without annotation in circBase were considered novel circRNAs.

### 4.5. Cell Culture and Loss-of-Function Studies

The 697 cell line, established from the bone marrow of a 12-year-old boy with ALL, was cultured in RPMI 1640 media (Wisent BioProducts, Montreal, QC, Canada) supplemented with fetal calf serum (Wisent Bio Products) and 100 units/mL of penicillin–streptomycin (Wisent BioProducts) at 37 °C with 5% CO_2_, in a humidified atmosphere. Localization of *circBARD1* was determined by RT-qPCR on the nuclear and cytoplasmic fraction isolated using the Paris kit (Thermo Fisher Scientific, Waltham, MA, USA) with the GAPDH and U6 transcript as markers for cytoplasmic and nuclear fraction, respectively. Gene knockdowns of both *NUDT21* and *circBARD1* were performed using the CRISPR-Cas13d system as previously described [33,53,54]. Briefly, stable cell lines expressing Cas13d protein (plasmid kindly provided by Dr. Martin Sauvageau, Addgene, Watertown, MA, USA, #109049), were generated using lentiviral expression vectors co-expressing GFP and sorted by FACS. To improve silencing efficiency, two 23-nucleotide gRNAs recognizing distinct *NUDT21* exons and three 30-nucleotide gRNAs spanning the blacksplicing junction of *circBARD1* were designed and cloned into a vector allowing their expression from a unique promoter (Cas13d-gNUDT21 and Cas13d-gcircBARD1). A non-targeting (NT) gRNA was used as the control (Cas13d-NT). The selection of cells expressing gRNAs was performed by adding puromycin (1 μg/mL) (Sigma-Aldrich, Burlington, MA, USA). The expression of *NUDT21* and both circular and parental gene forms of circRNA candidates were analyzed by RT-qPCR in triplicate. All primers and gRNAs sequences are listed in Appendix A.

### 4.6. In Vitro Assessment of circRNAs Silencing on Cancer Phenotypes

(A) Proliferation. Cells were inoculated in triplicate at 1.5 × 10^4^ cells/mL in 3 mL in a 12-well plate and were counted daily over 6 days with a Z1 Coulter particle counter (Beckman-Coulter, Mississauga, ON, Canada). (B) Migration: Cells were seeded in the upper chamber of polycarbonate filters with a pore size of 5 μm (BD Biosciences, Bedford, MA, USA) and placed in a 12-well plate filled with serum-free RPMI-1640 medium supplemented with 100 nM SDF-1 (Sigma-Aldrich) in the lower chamber. After overnight incubation, cells in the lower chamber were counted with a Z1 Coulter particle counter (Beckman-Coulter). (C) Oxidative stress induction: Cells were seeded at 3 × 10^4^ and treated with increasing concentrations of hydrogen peroxide (H_2_O_2_). Cell viability was measured by CellTiter Glo 24 h after H_2_O_2_ addition. (D) Protein phosphorylation of ERK/MAP kinase 1/2, Akt, STAT3, JNK, p70S6 kinase, NF-kB, STAT5A/B, CREB, and p38 was determined using the Luminex 2000, with 9-plex Multi-Pathway Magnetic Bead Panel (Millipore #46-680MAG, Amsterdam, Netherlands) following the manufacturer’s protocol. Cell lysate (Millipore #47-210) was used as a negative control and GAPDH beads (Millipore #46-667MAG) were added to correct for protein loading. Experiments were performed in triplicate.

## Figures and Tables

**Figure 1 ijms-25-01477-f001:**
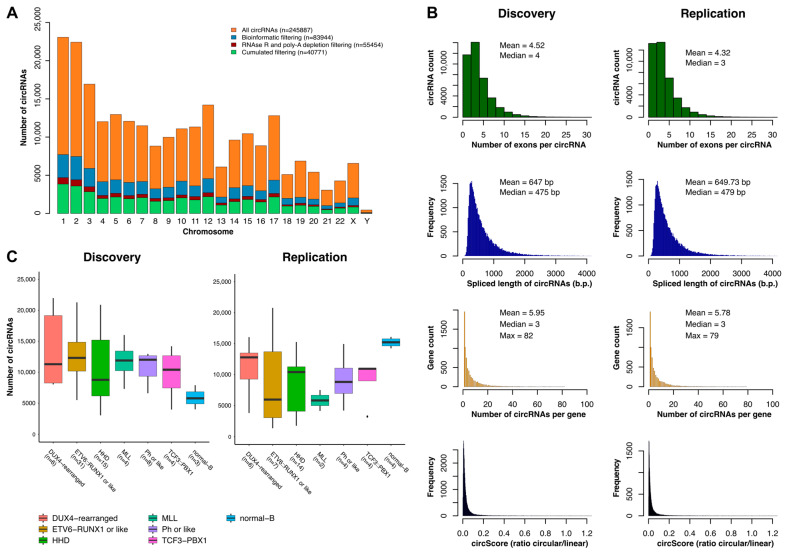
Overview of the circRNAome in childhood B-ALL. (**A**) Number of circRNAs identified per chromosome before and after bioinformatic and biochemical filtering. (**B**) The mean size, average number of exons as well as the mean number of circRNA isoforms per gene of circRNAs identified. (**C**) Number of circRNAs identified in every ALL subtype and controls.

**Figure 2 ijms-25-01477-f002:**
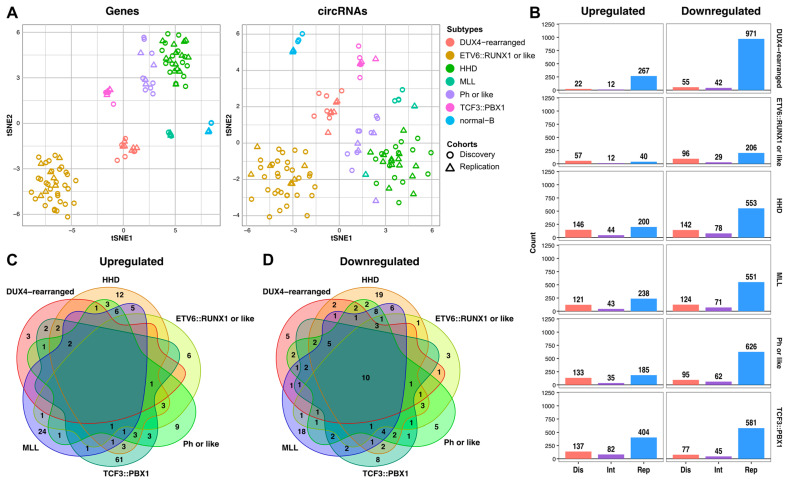
Characterization of circRNA profiles. (**A**) Unsupervised clustering analysis (*n* = 112) using t-SNE shows the stratification of childhood ALL subtypes using the 500 most variable genes and circRNAs. (**B**) Total number of circRNAs differentially expressed in discovery (Dis) and replication (Rep) cohorts and integration of the 2 cohorts (Int). (**C**) Venn diagram of overexpressed circRNAs for each subtype. (**D**) Venn diagram of underexpressed circRNAs for each subtype.

**Figure 3 ijms-25-01477-f003:**
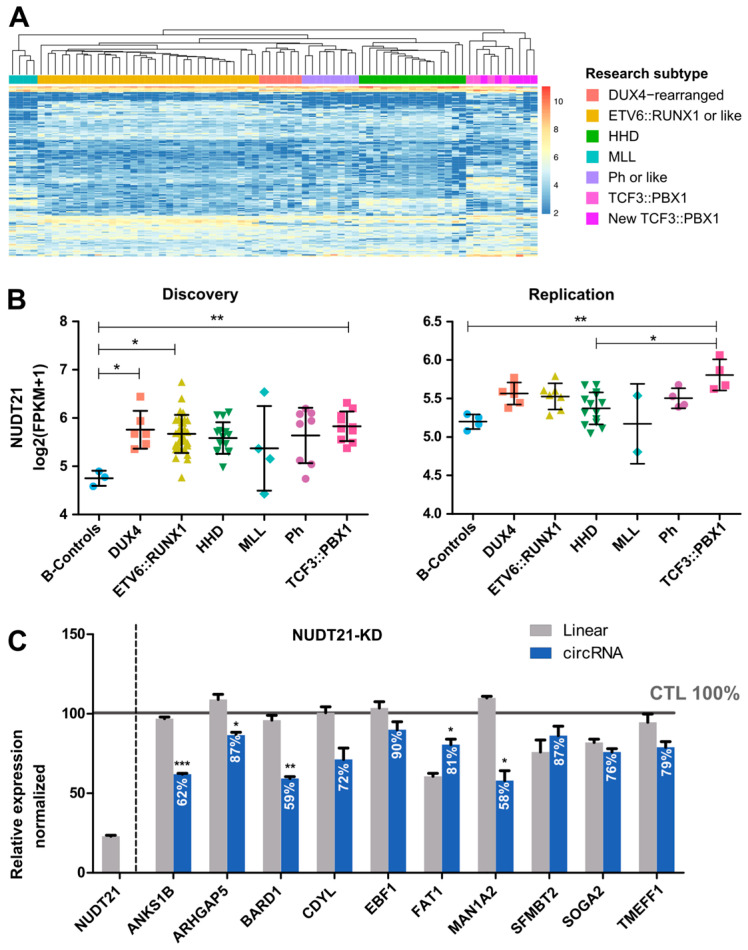
CircRNAome in *TCF3::PBX1* ALL. (**A**) Hierarchical clustering using the circRNA signature (156 overexpressed and 134 underexpressed circRNAs) to cluster B-ALL subtypes and new patients with *TCF3::PBX1* ALL. (**B**) FPKM values of NUDT21 in the discovery and validation cohorts were extracted from the transcriptome data and log transformed. Only significant *p*-values (<0.05) are indicated (ANOVA test, corrected for multiple testing with Bonferroni). (**C**) RT-qPCR analysis of 10 circRNAs and their cognate pre-mRNAs in 697 control cells and NUDT21-KD cells. Experiments were performed in triplicate. Comparisons assessed by *t*-test; * *p*-value < 0.05; ** *p*-value < 0.005; *** *p*-value < 0.0005.

**Figure 4 ijms-25-01477-f004:**
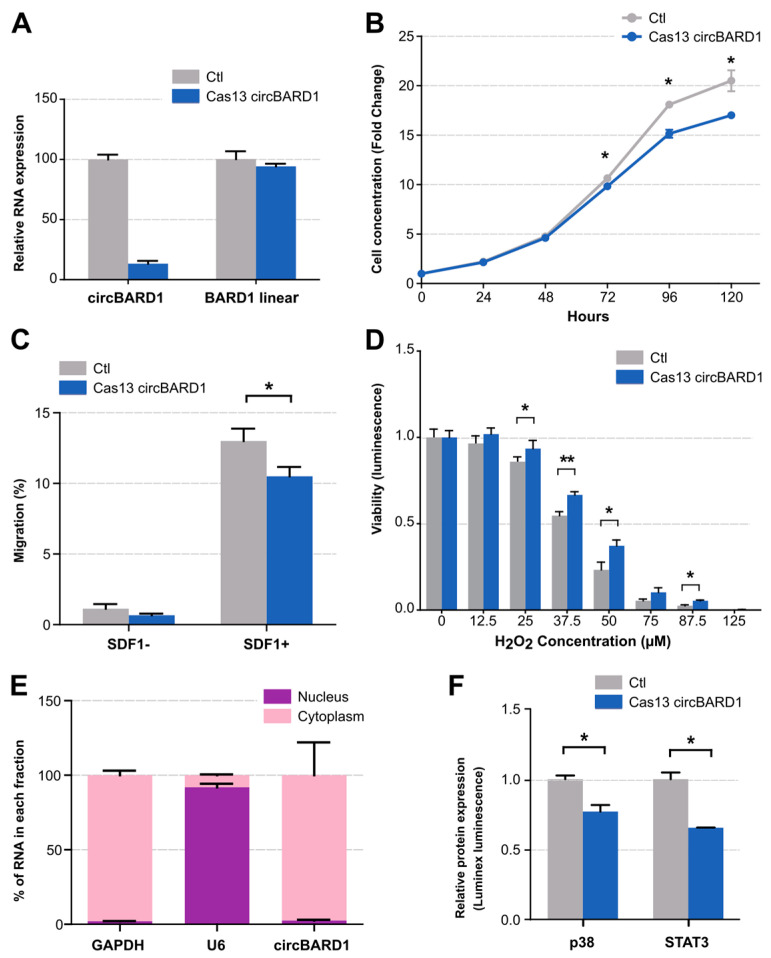
Functional characterization of *circBARD1* by loss-of-function studies. (**A**) Expression of *circBARD1* and the linear *BARD1* transcript in *circBARD1* silenced cell line vs. control. Effects of the silencing of *circBARD1* on (**B**) migration, (**C**) proliferation, and (**D**) oxidative stress induction. (**E**) Expression of *circBARD1* in the cytoplasmic fraction of 697 cells. (**F**) Effects of the silencing of *circBARD1* on protein phosphorylation. Experiments were performed in triplicate. Comparisons assessed by *t*-test; * *p*-value < 0.05; ** *p*-value < 0.005.

**Table 1 ijms-25-01477-t001:** Patients’ characteristics.

	Discovery	Replication
No. of individuals	74	37
Mean age ± SE, y	6.9 ± 4.4	6.3 ± 4.4
Sex *		
Males, *n* (%)	34 (45.9)	18 (48.6)
Females, *n* (%)	38 (51.4)	19 (51.4)
Genetic alterations		
Hyperdiploid, *n* (%)	15 (20.3)	14 (34.1)
*ETV6::RUNX1*, *n* (%)	31 (41.9)	7 (17.1)
*MLL*, *n* (%)	4 (5.4)	2 (4.9)
*BCR::ABL*, *n* (%)	8 (10.8)	4 (9.8)
*TCF3::PBX1*, *n* (%)	10 (13.5)	4 (9.8)
*DUX4*-rearranged, *n* (%)	6 (8.1)	6 (14.6)

SE: Standard Deviation; * No data for one patient in discovery cohort.

## Data Availability

The sequencing datasets generated and/or analyzed in the current study will be available in GEO under accession GSE206336.

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
