# Peer review of "CircRNAome of Childhood Acute Lymphoblastic Leukemia: Deciphering Subtype-Specific Expression Profiles and Involvement in TCF3::PBX1 ALL"

_ijms, 2024, doi:10.3390/ijms25031477_

Round 1

Reviewer 1 Report

Comments and Suggestions for Authors

The authors have studied different molecular subtypes of Childhood B-cell acute lymphoblastic leukemia (B-ALL) and found that TCF3-PBX1 subtype was the one with the highest number of unique and overexpressed circRNAs, and the circRNA signature could effectively discriminates new patients with TCF3-PBX1 subtype from others. In an attempt to identify the reason of this difference the authors focussed on the expression levels of ten genes involved in circRNA biogenesis and found that only one NUDT21 may contribute to this circRNA enrichment in TCF3-PBX1 ALL. To further explore the role of NUDT21 in circRNA biogenesis, they used the CRISPR-199 Cas13d system to silence NUDT21 in 697 cells and analyzed the expression levels of ten circRNA candidates and their corresponding parental genes by RT-qPCR. They showed that for 3 out of the 10 candidates (circANKS1B, circBARD1 and circMAN1A2), silencing NUDT21 decreased circRNA expression by 40%. At this stage they performed a functional validation of only circBARD1 and concluded that it might be involved in leukemogenesis through the activation of p38 via hsa-miR-153-5p. The results can be of interest in the field although their main weakness is the low number of TCF3-PBX1 ALL patients analyzed in both the discovery and replication cohort (n=4). Moreover, it remains unclear if the highest number of unique and overexpressed circRNAs in the TCF3-PBX1 subtype can be justified only by the dysregulation of NUDT21 and what are the causes of this specific dysregulation observed only in TCF3-PBX1 subtype.

Author Response

We thank the reviewer for their constructive comments which helped strengthen this manuscript. Taking their suggestions into careful consideration, we made the following changes to address their comments. Changes in the manuscript are  highlighted in gray.

Reviewer #1:

The authors have studied different molecular subtypes of Childhood B-cell acute lymphoblastic leukemia (B-ALL) and found that TCF3-PBX1 subtype was the one with the highest number of unique and overexpressed circRNAs, and the circRNA signature could effectively discriminates new patients with TCF3-PBX1 subtype from others. In an attempt to identify the reason of this difference the authors focussed on the expression levels of ten genes involved in circRNA biogenesis and found that only one NUDT21 may contribute to this circRNA enrichment in TCF3-PBX1 ALL. To further explore the role of NUDT21 in circRNA biogenesis, they used the CRISPR-199 Cas13d system to silence NUDT21 in 697 cells and analyzed the expression levels of ten circRNA candidates and their corresponding parental genes by RT-qPCR. They showed that for 3 out of the 10 candidates (circANKS1B, circBARD1 and circMAN1A2), silencing NUDT21 decreased circRNA expression by 40%. At this stage they performed a functional validation of only circBARD1 and concluded that it might be involved in leukemogenesis through the activation of p38 via hsa-miR-153-5p.

Question #1: The results can be of interest in the field although their main weakness is the low number of TCF3-PBX1 ALL patients analyzed in both the discovery and replication cohort (n=4).

The frequency of TCF3::PBX1 subtype in childhood B-ALL cases is around 7% (Lilljebjörn H and Fioretos T, Blood, 2017). While TCF3::PBX1 patients are underrepresented in the discovery cohort (5.9%), patients with this translocation comprise 10.8% of the replication cohort and account for 7.6% of cases when considering the whole study. Moreover, we included six additional patients with TCF3::PBX1 subtype as part of the discovery cohort to perform a more comprehensive analysis, increasing the representation of this subtype to 12.6% in the overall study. These analyses included hierarchical clustering, differential expression analysis, and correlation analysis. Hierarchical clustering using the 156 overexpressed and 134 underexpressed circRNAs identified revealed that our circRNA signature effectively discriminates new patients with TCF3::PBX1 subtype from others (Figure 3A). Moreover, differential expression analysis between TCF3::PBX1 and other B-ALL subtypes revealed a higher number of circRNAs deregulated in TCF3::PBX1 patients (Figure S4, Table S8). We realized that we did not include these additional patients in Table 1, featuring patient´s characteristics, we rectified this omission and clarified this point in the methods section.

It is crucial to note the inherent challenge associated with the low incidence of patients diagnosed with the TCF3::PBX1 subtype, a challenge that is increased by the difficulty  of adding public datasets (for example TARGET) that are more or less compatible with circRNA analysis due to the poly(A) selection step used during library preparation, thus causing the depletion of circRNAs. Despite this limitation, the number of patients analyzed in this study stand at the higher end when compared to other investigations profiling different subtypes in childhood leukemia (Buratin et al., Blood Advances 2020, Tretti Parenzan et al., Blood Advances 2023)

Question #2: Moreover, it remains unclear if the highest number of unique and overexpressed circRNAs in the TCF3-PBX1 subtype can be justified only by the dysregulation of NUDT21 and what are the causes of this specific dysregulation observed only in TCF3-PBX1 subtype

Advances in the understanding of circRNA biogenesis suggested that a broad range of RBPs are involved in circRNA formation through various mechanisms (Conn SJ et al., Cell 2015, Fei et al., Proc Natl Acad Sci U S A. 2017). Recent discoveries also highlight m6A modifications as significant contributors of circRNA biogenesis (Tang et al., Cell Res. 2020; Di Timoteo G Cell Rep 2020). In addition, epigenetic changes within gene bodies and histone modifications affect alternative splicing and may also have a direct impact on circRNA biogenesis (Kristensen et al., Nature Reviews, 2019). While we only focused on RBPs and showed that NUDT21 as an RBP is contributing to this dysregulation, the circRNA profiles observed in TCF3::PBX1 can arise from  multiple alteration of circRNA biogenesis and the factors involved in this process. We acknowledge this limitation in the discussion section: “Finally, only a few RNA-binding factors involved in circRNA biogenesis have been discovered (36). Here, we focused on the involvement of RBPs, but recent findings suggest that m6A modifications (37) and epigenetic changes within histones and genes (8) may also have a direct impact on circRNA biogenesis. Further studies on the mechanisms of circRNA dysregulation could help elucidate the whole picture of aberrant circRNA expression observed in TCF3::PBX1 patients.

Reviewer 2 Report

Comments and Suggestions for Authors

The study by Gutierrez-Camino et al. analyzed circular RNA in pre-B-cell leukemia. They identified PBC-ALL subtype TCF3::PBX1 showing conspiciously high levels of circRNA. Furthermore, they recognized NUDT21 overexpression in this subtype and demonstrated that NUDT21 activated generation of particular circRNAs including circBARD1. CircBARD1 in turn mediated activation of p38. Thus, htey comprehensively analyzed this novel type of molecules in a subset of ALL and showed their functional relevance. The study is well performed and presented. However, I have two minor points to be corrected:

1. The indicated source of used cell lines is wrong since the ATCC does not provide SEM, NALM-6, 697 and GM12878.

2. Please use the actual nomenclature to describe fusion genes (TCF3::PBX1).

Author Response

We thank the reviewer for their constructive comments which helped strengthen this manuscript. Taking their suggestions into careful consideration, we made the following changes to address their comments. Changes in the manuscript are  highlighted in gray.

Reviewer #2:

The study by Gutierrez-Camino et al. analyzed circular RNA in preB-cell leukemia. They identified PBC-ALL subtype TCF3::PBX1 showing conspiciously high levels of circRNA. Furthermore, they recognized NUDT21 overexpression in this subtype and demonstrated that NUDT21 activated generation of particular circRNAs including circBARD1. CircBARD1 in turn mediated activation of p38. Thus, htey comprehensively analyzed this novel type of molecules in a subset of ALL and showed their functional relevance. The study is well performed and presented. However, I have two minor points to be corrected:

Question #3: The indicated source of used cell lines is wrong since the ATCC does not provide SEM, NALM-6, 697 and GM12878.

Thank you for pointing out this mistake. In the new version of the manuscript, we clarified that SEM, NALM-6 and 697 were purchased from DSMZ-German Collection of Microorganisms and Cell Cultures GmbH, Braunschweig, Germany and GM12878 from Coriell Institute for Medical Research, NJ, USA.

Question #4: Please use the actual nomenclature to describe fusion genes (TCF3::PBX1).

We have addressed the nomenclature for fusion genes throughout the text and figures.

Reviewer 3 Report

Comments and Suggestions for Authors

In this manuscript, Gutierrez-Camino et al. defined specific signatures of circRNA that stratify childhood B-ALL subtypes, identifying mainly the TCF3-PBX1 group.

Although the topic is potentially interesting, there are several points that need improvement, which are reported below.

In the introduction the mechanism of action of circRNAs needs to be explained in more detail.

The difference between the discovery and replication cohorts is missed and needs to be better explained.

Justify the different number of patients for the different subtypes: is it representative of the relative frequency in patients?

The choice to use two different types of control cells in the different cohorts is not clear. It would be better to make the controls homogeneous, perhaps using half of the cord blood samples and half of the peripheral blood samples for both cohorts.

Report the definition of CIRCscore also in the results paragraph where it is mentioned for the first time (line 116) or at least report that the definition is in materials and methods.

Describe the targets of circRNAs important in B-ALL.

Lines 163: replace the term "boys" with "males".

When talking about the 167 cell line, describe what type of cells they are (e.g. derivation). In this regard, I suggest validating the results obtained in more than one cell line (at least 3).

Line 220: there are no references relating to the increase in ROS in tumor cells (e.g. doi: 10.1038/s41416-019-0651-y, 10.1016/j.metabol.2023.155747).

Line 256: the number of unique circRNA differentially expressed is wrong (156+134=290 not 289).

Round 2

Reviewer 3 Report

Comments and Suggestions for Authors

Thank to the Authors have replied to the comments.

The manuscript is now suitable for the publication.